# Translational control of nociception via 4E-binding protein 1

Arkady Khoutorsky[1,2†], Robert P Bonin[3†], Robert E Sorge[4,5†], Christos G Gkogkas[1,6], Sophie Anne Pawlowski[5,7], Seyed Mehdi Jafarnejad[1,2], Mark H Pitcher[8], Tommy Alain[1,2], Jimena Perez-Sanchez[3], Eric W Salter[3], Loren Martin[4,5], Alfredo Ribeiro-da-Silva[5,7], Yves De Koninck[3,5,7,9], Fernando Cervero[5,10], Jeffrey S Mogil[4,5*], Nahum Sonenberg[1,2*]

[1]Department of Biochemistry, McGill University, Montréal, Canada; [2]Rosalind and Morris Goodman Cancer Research Centre, McGill University, Montréal, Canada; [3]Unité de neurosciences cellulaires et moléculaire, Institut universitaire en santé mentale de Québec, Québec, Canada; [4]Department of Psychology, McGill University, Montréal, Canada; [5]Alan Edwards Centre for Research on Pain, McGill University, Montréal, Canada; [6]Centre for Integrative Physiology and The Patrick Wild Centre, University of Edinburgh, Edinburgh, United Kingdom; [7]Department of Pharmacology and Therapeutics, McGill University, Montreal, Canada; [8]National Center for Complementary and Alternative Medicine, National Institutes of Health, Porter Neuroscience Research Center, Maryland, United States; [9]Department of Psychiatry and Neuroscience, Université Laval, Québec, Canada; [10]Anesthesia Research Unit, McGill University, Montreal, Canada

*For correspondence: jeffrey.mogil@mcgill.ca (JSM); nahum.sonenberg@mcgill.ca (NS)

†These authors contributed equally to this work

**Abstract** Activation of the mechanistic/mammalian target of rapamycin (mTOR) kinase in models of acute and chronic pain is strongly implicated in mediating enhanced translation and hyperalgesia. However, the molecular mechanisms by which mTOR regulates nociception remain unclear. Here we show that deletion of the eukaryotic initiation factor 4E-binding protein 1 (4E-BP1), a major mTOR downstream effector, which represses eIF4E activity and cap-dependent translation, leads to mechanical, but not thermal pain hypersensitivity. Mice lacking 4E-BP1 exhibit enhanced spinal cord expression of neuroligin 1, a cell-adhesion postsynaptic protein regulating excitatory synapse function, and show increased excitatory synaptic input into spinal neurons, and a lowered threshold for induction of synaptic potentiation. Pharmacological inhibition of eIF4E or genetic reduction of neuroligin 1 levels normalizes the increased excitatory synaptic activity and reverses mechanical hypersensitivity. Thus, translational control by 4E-BP1 downstream of mTOR effects the expression of neuroligin 1 and excitatory synaptic transmission in the spinal cord, and thereby contributes to enhanced mechanical nociception.

## Introduction

De novo gene expression induced by noxious stimuli markedly contributes to the development of pain hypersensitivity (i.e., allodynia and hyperalgesia). Regulation of gene expression at the level of translation enables the cell to rapidly change its proteome by modulating the rate of mRNA translation without altering mRNA levels (*Sonenberg and Hinnebusch, 2009*). Upregulation of mRNA translation via activation of the mechanical/mammalian target of rapamycin (mTOR) by noxious stimuli has been proposed to sensitize primary nociceptors and spinal circuits (*Bogen et al., 2012*; *Ferrari et al., 2013*; *Jimenez-Diaz et al., 2008*; *Melemedjian et al., 2010*; *Obara and Hunt, 2014*;

**eLife digest** Despite the unpleasant feeling it causes, pain is necessary for survival as it helps individuals to avoid objects, environments and situations that cause damage to their body. However, millions of people experience long-lasting "chronic" pain, or are hypersensitive to pain. There are few treatments available for these conditions, but these treatments do not work well for the majority of patients, and can have serious side effects. To develop new treatments, researchers must first better understand how chronic pain develops.

Pain is transmitted to the brain in the form of electrical signals "fired" along nerve fibers. Different nerves transmit information about different types of pain: for example, pain caused by a sharp object pressed against the skin activates a different set of neurons to those activated when touching something dangerously hot. Studies in mice have suggested that a protein called mTOR that is found inside neurons is important for them to fire pain signals. However, it is not clear exactly how mTOR contributes to pain signaling, although it is known to affect the activities of several other proteins in neurons.

One protein that mTOR affects the activity of is called 4E-BP1. Now, Khoutorsky, Bonin, Sorge et al. show that mice that lack 4E-BP1 behave in ways that suggest they are hypersensitive to poking or pinching sensations. However, the mice did not show hypersensitivity when they touched a hot surface. Further investigation revealed that the neurons in the spinal cord of mice that lack 4E-BP1 produce abnormally high amounts of a molecule called neuroligin 1, which makes the neurons more likely to fire and thus signal pain.

Khoutorsky, Bonin, Sorge et al. found that treating mice that lack 4E-BP1 with a compound that reduces neuroligin 1 production causes their neurons to fire more normally. This also reduces the animals' apparent signs of hypersensitivity to pressure on their skin. It will be important in future studies to identify additional targets of 4E-BP1 in the spinal cord that could contribute to increased mechanical sensation, and also to study the role of 4E-BP1 in peripheral nerves.

*Price and Geranton, 2009*). mTOR is an evolutionarily conserved serine/threonine kinase that controls cell homeostasis through key molecular processes including translation, lipid biogenesis, autophagy, and cytoskeleton organization (*Shimobayashi and Hall, 2014*). mTOR is the catalytic subunit of two structurally and functionally distinct multiprotein complexes, mTORC1 and mTORC2 (*Lipton and Sahin, 2014*). mTORC1 is defined by the protein raptor and is sensitive to rapamycin, while mTORC2 is defined by the protein rictor and is rapamycin insensitive. mTORC1 is a key regulator of translation, whereas mTORC2 controls the actin cytoskeleton (*Jacinto et al., 2004*; *Sarbassov et al., 2004*). The mTORC1 pathway is activated in primary nociceptors and superficial dorsal horn neurons in rodent models of inflammatory pain (*Jiang et al., 2013*; *Liang et al., 2013*; *Norsted Gregory et al., 2010*; *Xu et al., 2011*), bone cancer-induced pain (*Shih et al., 2012*), neuropathic pain (*Zhang et al., 2013*), and in response to repeated morphine administration (*Xu et al., 2015*; *Xu et al., 2014*). The functional role of mTORC1 activation has been studied using the mTORC1 specific inhibitor, rapamycin, and its derivatives (rapalogues). Systemic or intrathecal (i.t.) administration of rapalogues does not affect acute responses to mechanical and thermal stimuli (*Geranton et al., 2009*; *Xu et al., 2014*), but it reduces nocifensive behaviors, and normalizes mechanical hypersensitivity in rodent models of inflammatory pain (*Asante et al., 2009*; *Jiang et al., 2013*; *Price and Geranton, 2009*; *Price et al., 2007*), bone cancer-induced pain (*Shih et al., 2012*), and neuropathic pain (*Asante et al., 2010*; *Cui et al., 2014*; *Zhang et al., 2013*). Taken together, these findings indicate that noxious stimuli-induced activation of mTORC1 plays an important role in the development of pain hypersensitivity. However, the cellular and molecular mechanisms mediating the effect of mTORC1 on nociception are not known.

All nuclear transcribed eukaryotic mRNAs harbor, at their 5′ end, the structure m7GpppN (where N is any nucleotide) termed 'cap', which facilitates ribosome recruitment to the mRNA. Recruitment of the ribosome to the mRNA is primarily regulated at the initiation step. A critical step of this process is the assembly of the eukaryotic initiation factor 4F (eIF4F) complex, which consists of eIF4E, the cap binding protein subunit, eIF4G, a large scaffold protein, and eIF4A, an RNA helicase that unwinds the mRNA 5′ UTR (untranslated region) secondary structure. Because eIF4E generally

exhibits the lowest expression level of all eukaryotic initiation factors, the cap-recognition step is rate-limiting for translation and a major target for regulation (*Sonenberg and Hinnebusch, 2009*). mTORC1 is a key regulator of translation initiation via phosphorylation of its downstream targets, eIF4E-binding proteins (4E-BPs) and p70S6 kinases (S6Ks). In their hypo-phosphorylated form, 4E-BPs compete with eIF4G for binding to eIF4E to repress eIF4F complex assembly, and thereby translation initiation. Following phosphorylation by mTORC1, 4E-BPs dissociate from eIF4E, allowing for eIF4F complex formation and activation of translation (*Gingras et al., 1996*). Three 4E-BP isoforms (1, 2, and 3) have similar functions, differing in their tissue distribution. 4E-BP3 expression is limited to few tissues, and is excluded from the nervous system. In the forebrain, 4E-BP2 is the major isoform, and 4E-BP2 knockout mice exhibit enhanced excitation (*Gkogkas et al., 2013*; *Ran et al., 2013*), alterations in synaptic plasticity, memory (*Banko et al., 2007*; *Banko et al., 2005*), and social behavior (*Gkogkas et al., 2013*). In the suprachiasmatic nucleus, 4E-BP1 controls circadian clock functions via regulation of vasoactive intestinal polypeptide (*Vip*) mRNA translation (*Cao et al., 2013*). In the pain pathway, 4E-BP1 is expressed in neurons in the dorsal horn of the spinal cord, and in peripheral nerves (*Jimenez-Diaz et al., 2008*; *Melemedjian et al., 2011*), whereas in the dorsal root ganglion (DRG) it was detected only in satellite glial cells (*Xu et al., 2010*).

Decreasing S6Ks activity by either pharmacological inhibitors or S6K1/2 null mutation unexpectedly led to mechanical allodynia via activation of ERK, a well-known pain sensitizing molecule (*Melemedjian et al., 2013*). Activation of ERK is caused by disinhibition of the S6K1/IRS-1/ERK negative feedback loop (*Carracedo et al., 2008*). Here, we investigated the role of the other major mTORC1 downstream effectors, 4E-BPs, in nociception using *Eif4ebp1 and 2* knockout (*Eif4epb$^{-/-}$*) mice. Removal of 4E-BP1 and 2 relieves eIF4E suppression and provides an opportunity to study cellular and molecular mechanisms underlying the impact of mTORC1, via 4E-BPs, on nociception. Previous work showed that intraplantar administration of two well-known sensitizers of nociceptors, IL-6 and NGF, upregulates eIF4F complex formation and cap-dependent translation in primary afferent neurons (*Melemedjian et al., 2010*). Peripheral inhibition of eIF4F with a specific inhibitor, 4EGI-1, completely blocked IL-6- and NGF-induced mechanical allodynia, supporting the idea that enhanced eIF4F complex formation and cap-dependent translation contribute to sensitization of primary afferents. However, the roles of eIF4F and 4E-BPs in the regulation of spinal circuits are unknown. Here we show that *Eif4ebp1$^{-/-}$*, but not *Eif4ebp2$^{-/-}$* mice exhibit mechanical (but not thermal) hypersensitivity, which can be rescued by intrathecal administration of 4EGI-1. Mechanical hypersensitivity is also elicited by downregulation of 4E-BP1 in the spinal cord dorsal horn. In *Eif4ebp1$^{-/-}$* mice, the excitatory synaptic input into spinal neurons is enhanced and the threshold for potentiation of electrically-evoked field post-synaptic potentials (fPSPs) in the superficial dorsal horn is lowered. The enhancement of excitatory synaptic transmission is mediated via increased synthesis of neuroligin 1, which promotes the excitatory synapse function. Reduction of neuroligin 1 levels in *Eif4ebp1$^{-/-}$* mice normalizes excitatory synaptic transmission, and partially reverses mechanical allodynia. Collectively, our data demonstrate that de-repression of eIF4E, via removal of 4E-BP1, contributes to mechanical allodynia through a neuroligin 1-mediated increase in the excitatory synaptic drive to the spinal circuit, and provides a new paradigm to explain the mechanisms by which mTORC1 controls nociception.

## Results

### 4E-BP1 ablation induces mechanical hypersensitivity

To study the role of 4E-BPs and eIF4F in nociception, we assessed mechanical and thermal sensation in *Eif4ebp1$^{-/-}$*, *Eif4ebp2$^{-/-}$*, and *Eif4ebp1$^{-/-}$/2$^{-/-}$* mice and wild-type controls. *Eif4ebp1$^{-/-}$* mice exhibited a marked increase in mechanical sensation in von Frey and tail clip tests (45% decrease in withdrawal threshold and 52% reduction in latency to attack tail clip, respectively, *Figure 1A*), whereas their thermal sensitivity was unaffected in both the radiant heat paw-withdrawal and hot-plate tests (*Figure 1B*). Additionally, in the chemical/inflammatory formalin test (20 µl, 0.5% formalin, intraplantar) *Eif4ebp1$^{-/-}$* mice showed higher levels of nocifensive behavior (*Figure 1C*) and increased c-Fos expression in the dorsal horn of the lumbar spinal cord (*Figure 1D*). In contrast, *Eif4ebp2$^{-/-}$* mice showed no mechanical (von Frey and tail clip tests, *Figure 1—figure supplement 1A*) or thermal pain phenotype (radiant heat paw-withdrawal and hot-plate tests, *Figure 1—figure supplement*

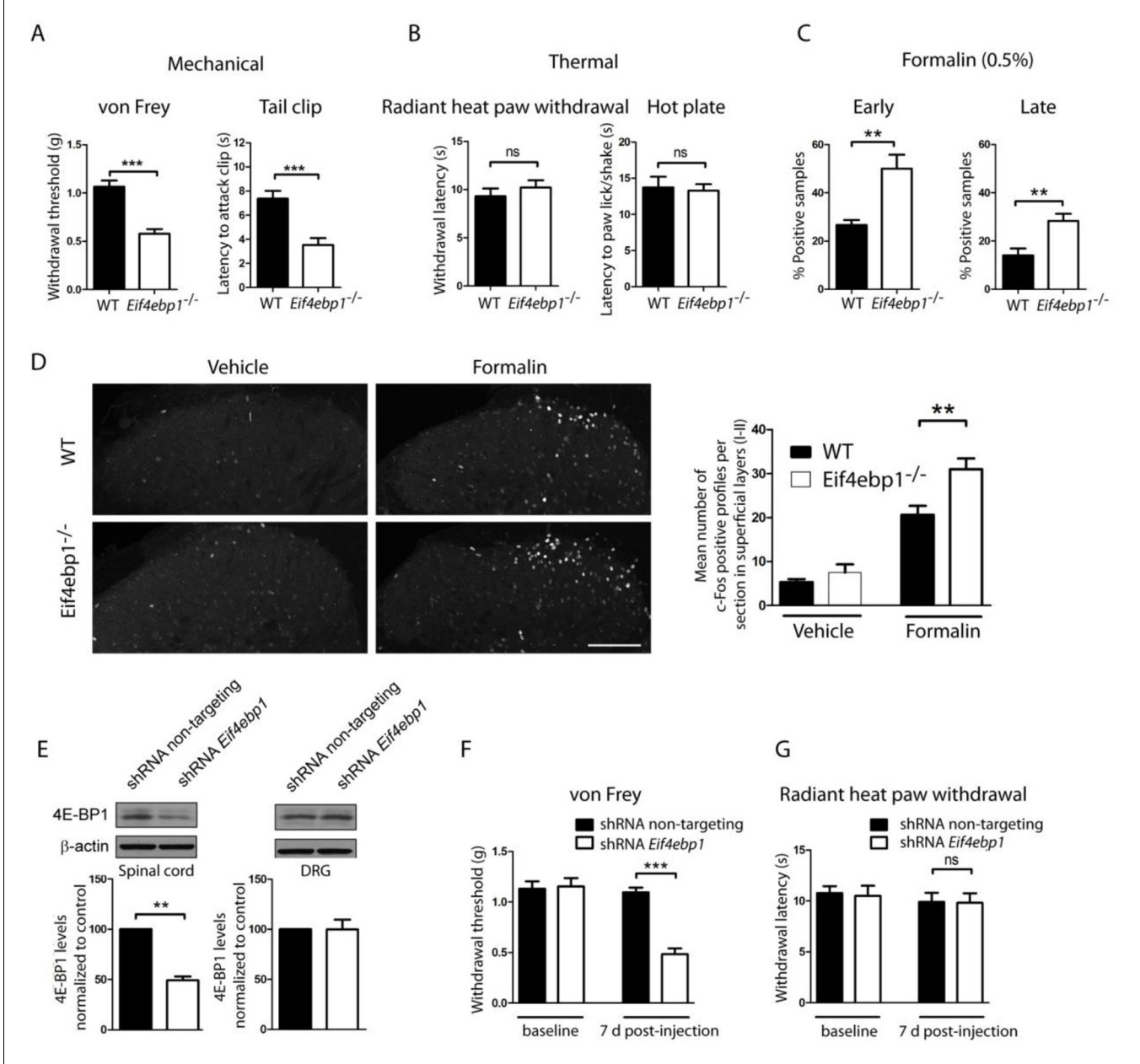

**Figure 1.** *Eif4ebp1⁻/⁻* mice exhibit mechanical hypersensitivity and increased formalin response. Mechanical pain sensitivity is increased in *Eif4ebp1⁻/⁻* mice as evident by decreased von Frey thresholds (**A**, n=8/genotype) and shortened latencies to attack tail clip (**A**, n=8/genotype), whereas thermal sensitivity is not altered (**B**, n=8/genotype). (**C**) *Eif4ebp1⁻/⁻* mice show more nocifensive (licking/shaking) behavior of the injected hind paw than their wild-type (WT) littermates in the formalin test (0.5%, 20 μl, n=8/genotype). Changes in paw weight, indicative of formalin-induced inflammation, were not different in *Eif4ebp1⁻/⁻* mice (not shown). (**D**) Intraplantar injection of formalin (0.5%, 20 μl) induced an enhanced upregulation of c-Fos (2 h post injection) in superficial layers (I-II) of the lumbar spinal cord of *Eif4ebp1⁻/⁻* as compared to WT mice (n=3 mice/group, 10 sections/mouse). (**E**) Western blot analysis of lysates prepared from the dorsal horn of the lumbar spinal cord and L3/L4 DRGs seven days post intraparenchymal dorsal horn injection of lentiviruses expressing shRNA against Eif4ebp1 as compared to non-targeting (scrambled) shRNA (n=3/condition). Mice injected with shRNA against *Eif4ebp1* exhibit a reduction in von Frey threshold 7 days post injection (**F**, n=9/condition), whereas thermal sensitivity is not altered (**G**, n=9/condition). Data are presented as mean ± SEM. *p<0.05, **p<0.01, ***p<0.001, ns–not significant by Student's *t*-test or *t*-test following repeated measures ANOVA. Scale bar: 100 μm. See also figure supplement 1 and 2.

The following figure supplements are available for Figure 1:

**Figure supplement 1.** *Eif4ebp1/2* DKO mice, but not *Eif4ebp2⁻/⁻*, mice exhibit mechanical hypersensitivity and enhanced formalin responding.

Figure 1. Continued

**Figure supplement 2.** No gross alterations in the dorsal horn of the *Eif4ebp1⁻/⁻* mice.

**Figure supplement 3.** Distribution of lentivirus-driven eGFP expression.

*1B*), and no alterations in formalin test behavior (*Figure 1—figure supplement 1C*). Mice lacking both 4E-BP1 and 4E-BP2 (*Eif4ebp1⁻/⁻/2⁻/⁻*) showed mechanical hypersensitivity and increased pain behavior in formalin test (*Figure 1—figure supplement 1D–F*), a phenotype identical to *Eif4ebp1⁻/⁻* mice. We did not detect any reorganization of the dorsal horn of *Eif4ebp1⁻/⁻* mice as assessed by neuronal marker NeuN (*Figure 1—figure supplement 2A*), projection of peptidergic (expressing substance P (SP) and calcitonin gene-related peptide (CGRP), *Figure 1—figure supplement 2B,C*) or non-peptidergic (expressing isolectin B4 (IB4), *Figure 1—figure supplement 2D*) C fibers, distribution of glial marker glial fibrillary acidic protein (GFAP, *Figure 1—figure supplement 2E*), and 5HT3A receptor (HTR3A, *Figure 1—figure supplement 2F*). To exclude the possibility that the phenotypes observed in *Eif4ebp1⁻/⁻* animals were the result of long-term developmental compensations and to test the role of 4E-BP1 selectively in the spinal cord, we knocked down *Eif4ebp1* in the dorsal horn of adult wild-type mice using an intraparenchymal injection of lentivirus expressing shRNA against *Eif4ebp1*. The virus-driven eGFP expression was restricted to the dorsal horn and showed mostly neuronal localization (*Figure 1—figure supplement 3*). Seven days after injection of *Eif4ebp1* shRNA into the dorsal horn of the lumbar spinal cord, mice showed reduced levels of 4E-BP1 protein in the lumbar spinal cord ($53 \pm 9\%$ reduction, *Figure 1E*), but not in DRGs, and exhibited mechanical, but not thermal hypersensitivity (*Figure 1F–G*). To study the distribution of 4E-BP1 and 4E-BP2, we performed western blot analysis of proteins prepared from DRG and superficial spinal cord, which showed the presence of 4E-BP1, but much less of the 4E-BP2 isoform (*Figure 1—figure supplement 2G*). Taken together, our data demonstrate that 4E-BP1 is the predominant functional isoform in the pain pathway and its downregulation in the spinal cord elicits mechanical hypersensitivity.

4E-BP1 suppresses eIF4F complex formation by competing with eIF4G for binding to eIF4E. Thus, removal of 4E-BP1 leads to increased levels of the eIF4F complex (*Gkogkas et al., 2013*; *Pause et al., 1994*; *Tahmasebi et al., 2014*). Therefore, we measured eIF4F complex levels in *Eif4ebp1⁻/⁻* mice using a cap-binding assay (see experimental procedures). As expected, in the lumbar spinal cord lysates, the amount of eIF4G associated with the cap-bound eIF4E was higher in *Eif4ebp1⁻/⁻* mice as compared to wild-type mice (*Figure 2A*). Next, we used an inhibitor of the eIF4F complex, 4EGI-1, which binds to eIF4E and inhibits its interaction with eIF4G. Daily i.t. administration of low doses of 4EGI-1 (10 µg) over three days normalized the enhanced binding of eIF4G to eIF4E in *Eif4ebp1⁻/⁻* mice (*Figure 2A*), and rescued the mechanical hypersensitivity and enhanced formalin-induced pain behavior in these mice (*Figure 2B–C*). These data indicate that the mechanical sensitivity and increased responses in the formalin test in *Eif4ebp1⁻/⁻* mice are a consequence of augmented eIF4F complex formation.

Next, we investigated the molecular and cellular mechanisms underlying the mechanical hypersensitivity of *Eif4ebp1⁻/⁻* mice. We focused on spinal mechanisms by which 4E-BP1 controls mechanical sensitivity. mTOR and 4E-BPs are strongly implicated in the regulation of synaptic transmission and synaptic plasticity in pyramidal hippocampal neurons (*Gkogkas et al., 2013*; *Ran et al., 2013*; *Richter and Klann, 2009*; *Santini et al., 2013*; *Weston et al., 2012*). Thus, we measured spontaneous miniature excitatory and inhibitory postsynaptic inputs into lamina II spinal neurons in spinal cord slices prepared from *Eif4ebp1⁻/⁻* and wild-type mice. The amplitude of miniature excitatory postsynaptic currents (mEPSCs) was increased in spinal neurons in *Eif4ebp1⁻/⁻* as compared to wild-type slices, whereas there was no statistically significant change in mEPSC frequency (*Figure 3A*). The amplitude and frequency of inhibitory postsynaptic currents (mIPSC) were increased in *Eif4ebp1⁻/⁻* mice (*Figure 3B*). In the spinal cord, fast inhibitory synaptic transmission is mediated via GABA and glycine receptors. Interestingly, while glycinergic mIPSC amplitude and frequency were enhanced, no alterations were measured in GABA$_A$-mediated synaptic transmission (*Figure 3C*). The increase in the mEPSC amplitude in lamina II spinal neurons of *Eif4ebp1⁻/⁻* was reversed by i.t. administration of 4EGI-1 (10 µg, daily for 3 days, *Figure 3D*), however, 4EGI-1 had no effect on the

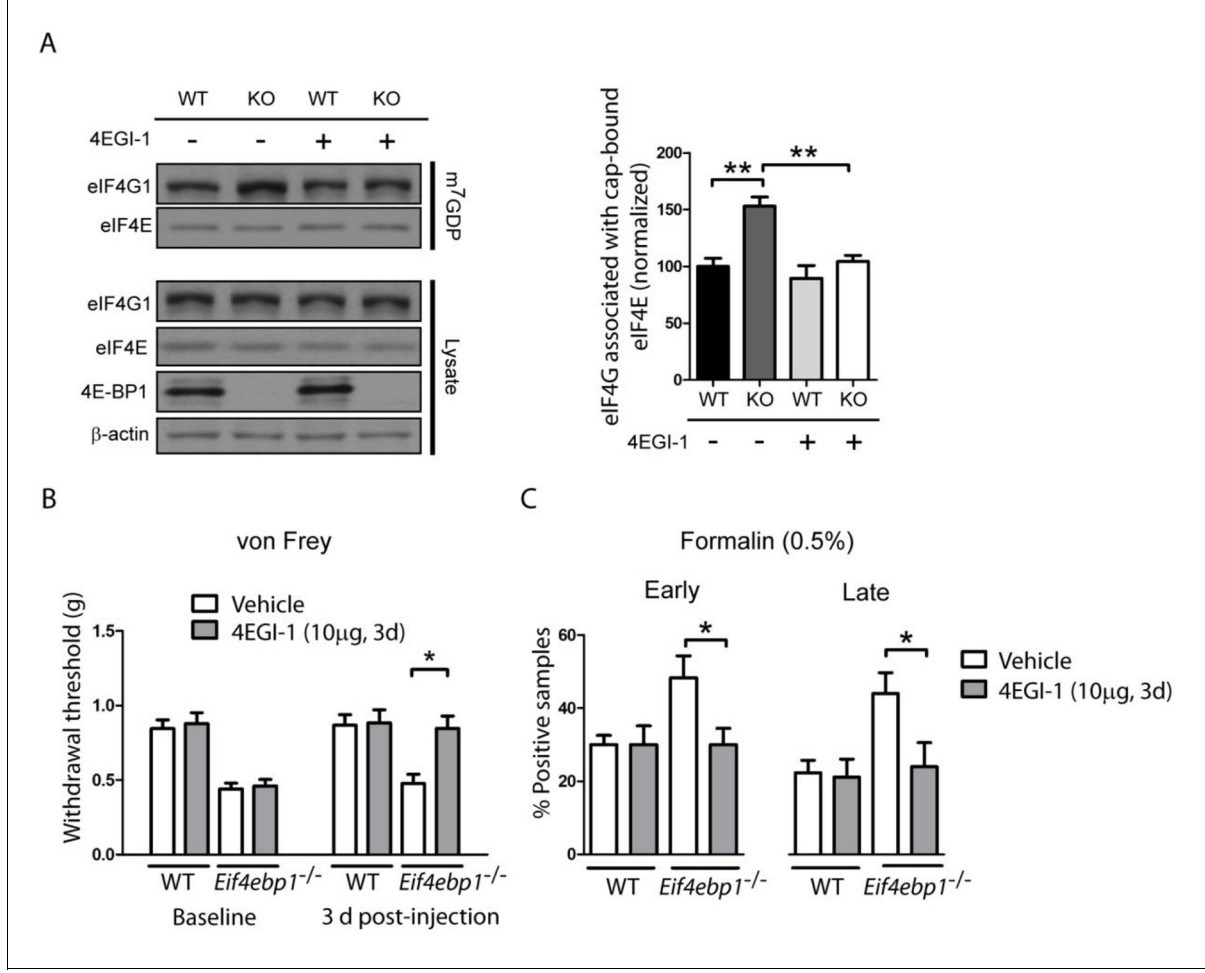

**Figure 2.** Increased levels of eIF4F complex in *Eif4ebp1*[-/-] mice cause mechanical hypersensitivity and increased formalin response. (A) Left, Immunoblot analysis of cap column pull-down proteins prepared from the spinal cord of *Eif4ebp1*[-/-] and wild-type mice (WT) treated with 4EGI-1 (10 μg, i.t. once a day for 3 days) or vehicle. Right, quantification of eIF4G1 in cap column pull-down material (n=4/genotype/drug). (B) Chronic 4EGI-1 treatment rescues mechanical hypersensitivity of *Eif4ebp1*[-/-] mice. von Frey threshold was measured prior to (baseline) and after 4EGI-1 chronic treatment (10 μg, i.t., once a day for 3 days, n=8/genotype/drug). (C) Chronic 4EGI-1 treatment normalizes the enhanced nocifensive (licking/shaking) behavior of *Eif4ebp1*[-/-] mice during the early (0–10 min post-formalin, 0.5%) and late phases (10–60 min post-formalin) of the formalin test (n=8/genotype/drug). Data are presented as mean ± SEM. *p<0.05, **p< 0.01 by Student's *t*-test following two-way (genotype x drug) ANOVA (in A, C) or following two-between (genotype, drug), one-within (repeated measures) ANOVA (in B).

increased amplitude and frequency of mIPSC (*Figure 3E*). The reasons for these differences will be addressed in the Discussion.

Additionally, we measured potentiation of fPSPs in *Eif4ebp1*[-/-] and wild-type lumbar spinal cords by stimulating dorsal roots (2 Hz) and recording extracellular fPSPs in the superficial layer of the spinal cord. The threshold for induction of potentiation was lowered in *Eif4ebp1*[-/-] as compared to wild-type mice, since potentiation was elicited with 40 s of stimulation in *Eif4ebp1*[-/-] spinal cords, whereas in wild-type spinal cords 60 s of stimulation were required to potentiate EPSPs (*Figure 3F*). The extent of potentiation achieved with 60 s of stimulation was 37.5% higher in *Eif4ebp1*[-/-] as compared to wild-type spinal cords. Together, the results demonstrate that basal synaptic transmission is enhanced, and the threshold for induction of synaptic potentiation is lowered in *Eif4ebp1*[-/-] mice.

Based on the effects of translational de-repression on synaptic physiology, as a consequence of 4E-BP1 depletion, we reasoned that translation of specific mRNAs, which regulate synaptic transmission, is affected. Consistent with previous reports, e.g. *Tahmasebi et al., 2014*, we found no differences in rates of general translation between wild-type and *Eif4ebp1*[-/-] mice, using the puromycin incorporation assay (*Figure 4A*), supporting the idea that alterations in translation of a small subset

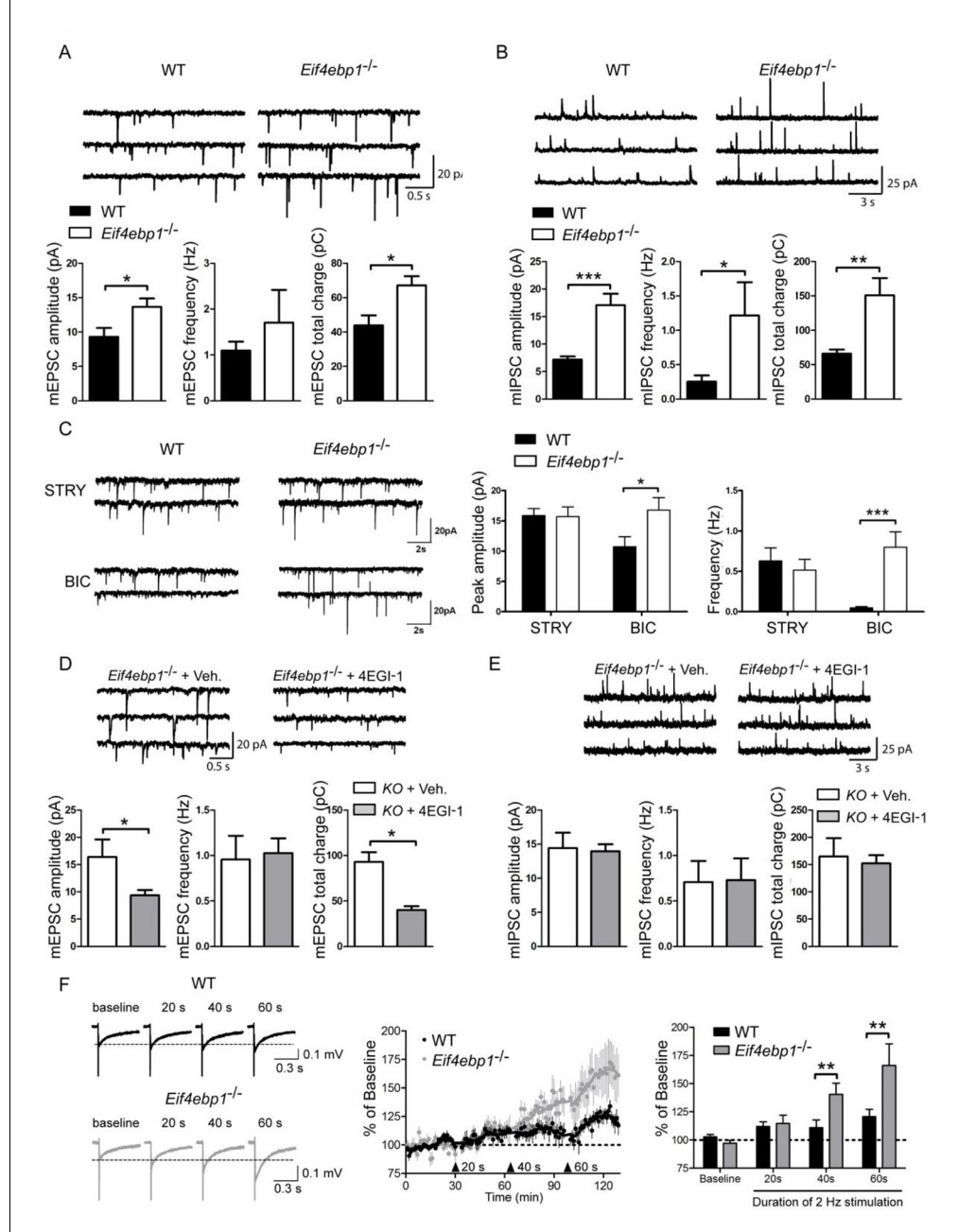

**Figure 3.** Excitatory and inhibitory synaptic transmissions are increased in the spinal cord of *Eif4ebp1*−/− mice. Representative traces (top) of mEPSCs (**A**) and mIPSCs (**B**) from lamina II neurons in acute lumbar spinal cord slices from *Eif4ebp1*−/− and wild-type mice (WT). Bar graphs (bottom) show mEPSC (**A**) and mIPSC (**B**) amplitude, frequency and synaptic total charge transfer (A: n=8 WT; n=7 *Eif4ebp1*−/−; B: n=9 WT; n=7 *Eif4ebp1*−/−). (**C**) To isolate GABA$_A$ or glycinergic mIPSCs, strychnine (STRY, glycine receptor antagonist, 1 μM) or bicuculline (BIC, GABA$_A$ receptor antagonist, 10 μM) were used, respectively. Left: representative traces of mIPSC in the presence of strychnine or bicuculline from *Eif4ebp1*−/− and WT slices. Right: bar graphs showing mIPSC amplitude and frequency in the presence of strychnine (n=7 WT; n=8 *Eif4ebp1*−/−) or bicuculline (n=9 WT; n=8 *Eif4ebp1*−/−). *Eif4ebp1*−/− mice were treated with 4EGI-1 (10 μg, daily for 3 days, i.t.) or vehicle, and the mEPSCs (**D**) and mIPSCs (**E**) were recorded from lamina II neurons (n=7 vehicle; n=11 4EGI-1). (**F**) Synaptic potentiation was elicited by stimulation of the dorsal root (2 Hz) for 20, 40 and 60 s and recording fEPSPs 125 μm from the dorsal surface of the spinal cord. Synaptic potentiation is induced in *Eif4ebp1*−/−, but not WT spinal cord preparation, by 40 s stimulation. Left: representative traces of fEPSPs, evoked by stimulation for the indicated length of time. Right: bar graph showing the summary of synaptic potentiation during the last 5 min prior to stimulation (n=7 WT; n=8 *Eif4ebp1*−/−). Data are presented as mean ± SEM. *p<0.05, **p<0.01, ***p<0.001 by Student's *t*-test (**A, B, D, E**), or by Student's *t*-test following two-way ANOVA (genotype x drug in C; genotype x duration in F).

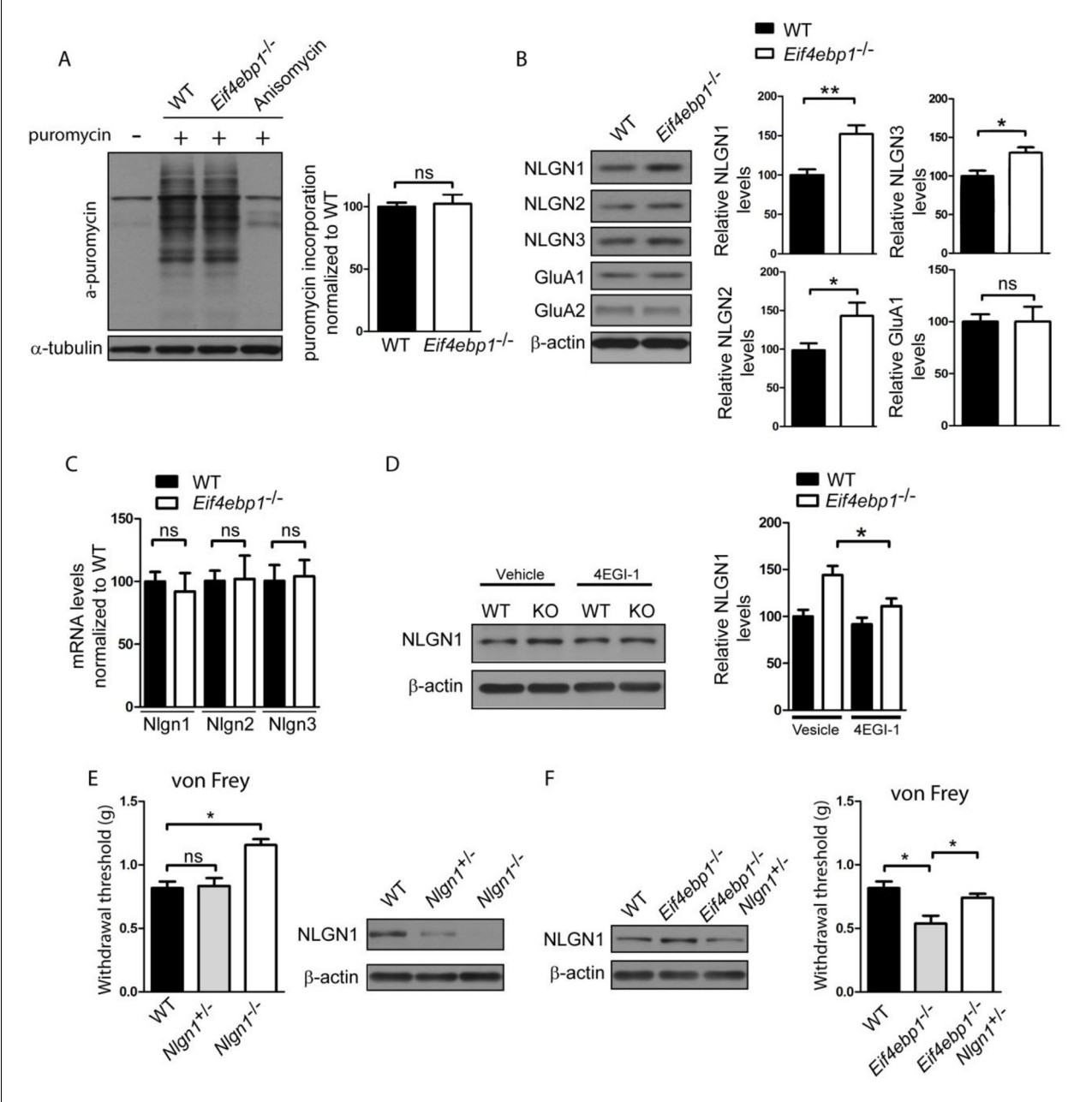

**Figure 4.** Enhanced expression of neuroligin 1 contributes to mechanical hypersensitivity of $Eif4ebp1^{-/-}$ mice. (A) General translation is not altered in $Eif4ebp1^{-/-}$ mice as assessed by puromycin incorporation (n=3/genotype). (B) Protein expression of neuroligin (NLGN) 1, 2 and 3 is increased in synaptosomes prepared from the dorsal horn of the lumbar spinal cord of $Eif4ebp1^{-/-}$ mice (n=5/genotype), whereas their mRNA levels are not changed (C). (D) Wild-type (WT) and $Eif4ebp1^{-/-}$ mice were treated with 4EGI-1 (10 μg, daily for 3 days, i.t.) or vehicle, and the protein levels of neuroligin 1 were measured in lysates prepared from the dorsal horn of the lumbar spinal cord using western blot analysis (n=5/condition). (E) $Nlgn1^{-/-}$ mice exhibit elevated von Frey thresholds, whereas $Nlgn1^{+/-}$ mice show no alteration in mechanical sensitivity (n=10 WT and $Nlgn1^{-/-}$; n=15 $Nlgn1^{+/-}$). (F) Deletion of one allele of $Nlgn1$ in $Eif4ebp1$ knockout mice ($Eif4ebp1^{-/-}$ / $Nlgn1^{+/-}$) normalizes neuroligin 1 protein levels in the dorsal horn of the lumbar spinal cord of $Eif4ebp1^{-/-}$ mice (left) and partially attenuates mechanical allodynia of $Eif4ebp1^{-/-}$ mice (right, n=10 wild-type, n=7 $Eif4ebp1^{-/-}$, n=14 $Eif4ebp1^{-/-}$ / $Nlgn1^{+/-}$). Data are presented as mean ± SEM. *p<0.05, **p<0.01, ns–not significant, by Student's *t*-test (A–C), Bonferroni post-hoc test following two-between (genotype, condition) one-within (repeated measures) ANOVA (E), or Bonferroni post-hoc test following one-way ANOVA (F).

of mRNAs engender nociceptive phenotypes in $Eif4ebp1^{-/-}$ mice. We previsouly investigated the mechanisms underlying the enhanced excitatory and inhibitory synaptic transmission in hippocampal

neurons of *Eif4ebp2*[-/-] mice (*Gkogkas et al., 2013*). 4E-BP2 is the major 4E-BP isoform in the hippocampus (*Banko et al., 2005*). Members of the family of post-synaptic adhesion proteins, neuroligins, were upregulated in the hippocampus of 4E-BP2 depleted mice, leading to changes in synaptic transmission. Specifically, translation of neuroligin (*Nlgn*) 1, 2 and 3 mRNAs was increased in the brain of *Eif4ebp2*[-/-] mice. Neuroligin 1 controls excitatory synapse formation, maturation, and function, whereas neuroligin 2 regulates inhibitory synaptic transmission (*Krueger et al., 2012*). Neuroligin 3 controls both excitatory and inhibitory synapses. An additional study has revealed a translational upregulation of α-amino-3-hydroxy-5-methyl-4-isoxazole propionic acid receptor (AMPA) receptor subunits, Glua1 and Glua2 in response to 4E-BP2 ablation (*Ran et al., 2013*). Protein levels of neuroligins (1–3) were increased in synaptosomes prepared from *Eif4ebp1*[-/-] spinal cords (*Figure 4B*), whereas their mRNA levels were unaltered (*Figure 4C*), suggesting that translation of neuroligin 1–3 mRNAs is increased in *Eif4ebp1*[-/-] mice. No differences in Glua1 and Glua2 protein expression were found in *Eif4ebp1*[-/-] spinal cords (*Figure 4B*).

In light of these results we hypothesized that the increased excitatory synaptic input into the spinal neurons and mechanical hypersensitivity in *Eif4ebp1*[-/-] mice are the result of increased neuroligin 1 expression, which is known to strongly promote excitatory synaptic transmission. The enhanced expression of neuroligin 1 in *Eif4ebp1*[-/-] spinal cord was normalized by 4EGI-1 treatment (10 μg, daily for 3 days, i.t. *Figure 4D*), consistent with our finding that 4EGI-1 normalizes the increased mEPSC amplitude in *Eif4ebp1*[-/-] spinal neurons (*Figure 3D*). To study the role of neuroligin 1 in nociception, we examined mice with a null mutation of neuroligin 1. *Nlgn1*[-/-] mice showed elevated von Frey thresholds (*Figure 4F*), indicating reduced mechanical sensation, whereas heterozygous mice (*Nlgn1*[+/-]) exhibited no change in their mechanical thresholds. Next, we normalized the enhanced expression of neuroligin 1 in *Eif4ebp1*[-/-] mice by removing one allele of *Nlgn1*. To this end, we generated *Eif4ebp1*[-/-]/ *Nlgn1*[+/-] double knockout mice and compared their mechanical sensitivity to *Eif4ebp1*[-/-] littermates. Deletion of one allele of *Nlgn1* in *Eif4ebp1*[-/-] mice normalized the increased neuroligin 1 expression (*Figure 4G*) and reversed mechanical sensitivity (*Figure 4G*). Taken together, our data demonstrate that elevated levels of neuroligin 1 contribute to the mechanical hypersensitivity of *Eif4ebp1*[-/-] mice.

Finally, we explored whether the increase in excitatory synaptic transmission in *Eif4ebp1*[-/-] mice is mediated via enhanced expression of neuroligin 1. Strikingly, deletion of one allele of *Nlgn1* in *Eif4ebp1*[-/-] mice (*Eif4ebp1*[-/-]/*Nlgn1*[+/-]) rescued the enhanced excitatory synaptic transmission (mEPSC amplitude and total charge transfer) in spinal neurons (*Figure 5A*), but had no effect on the inhibitory synapses (*Figure 5B*). No differences were found in input resistance and membrane capacitance of spinal neuron between the four genotypes examined (*Figure 5—figure supplement 1*). Thus, our data demonstrate that the enhanced expression of neuroligin 1 in mice lacking 4E-BP1 strengthens excitatory spinal synapses, and thereby contributes to mechanical hypersensitivity.

## Discussion

mTORC1 stimulates translation by relieving the 4E-BP-mediated block on eIF4F complex formation and cap-dependent translation. IL-6 and NGF-mediated enhancement of eIF4F complex formation play an important role in mediating mechanical nociceptive plasticity in primary afferent neurons (*Melemedjian et al., 2010*). However, the roles of eIF4F and 4E-BPs in regulation of spinal circuits remain unknown. Here, we show that de-repression of translation by removal of 4E-BP1 leads to enhanced eIF4F complex formation in the spinal cord, and to mechanical hypersensitivity and increased response to noxious chemical/inflammatory stimuli. This effect was observed in both *Eif4ebp1*[-/-] mice and in mice with selective downregulation of 4E-BP1 in the spinal cord dorsal horn. Normalization of eIF4F complex levels in *Eif4ebp1*[-/-] mice by intrathecal administration of 4EGI-1 reversed the exaggerated mechanical sensation, thus demonstrating that the enhanced eIF4F complex is the underlying cause for mechanical hypersensitivity in *Eif4ebp1*[-/-] mice.

eIF4F controls translation of a specific subset of mRNAs. Consistent with this and with previous reports (*Gkogkas et al., 2013*; *Tahmasebi et al., 2014*), we found that general translation is not altered in *Eif4ebp1*[-/-] mice (*Figure 4A*), indicating that upregulated translation of certain mRNAs contributes to the nociceptive phenotypes of these mice. Previous studies have documented that mRNAs in the family of neuroligin proteins are being translationally activated by eIF4F in the nervous system, such as in mice overexpressing eIF4E or lacking 4E-BP2 (*Gkogkas et al., 2013*). We

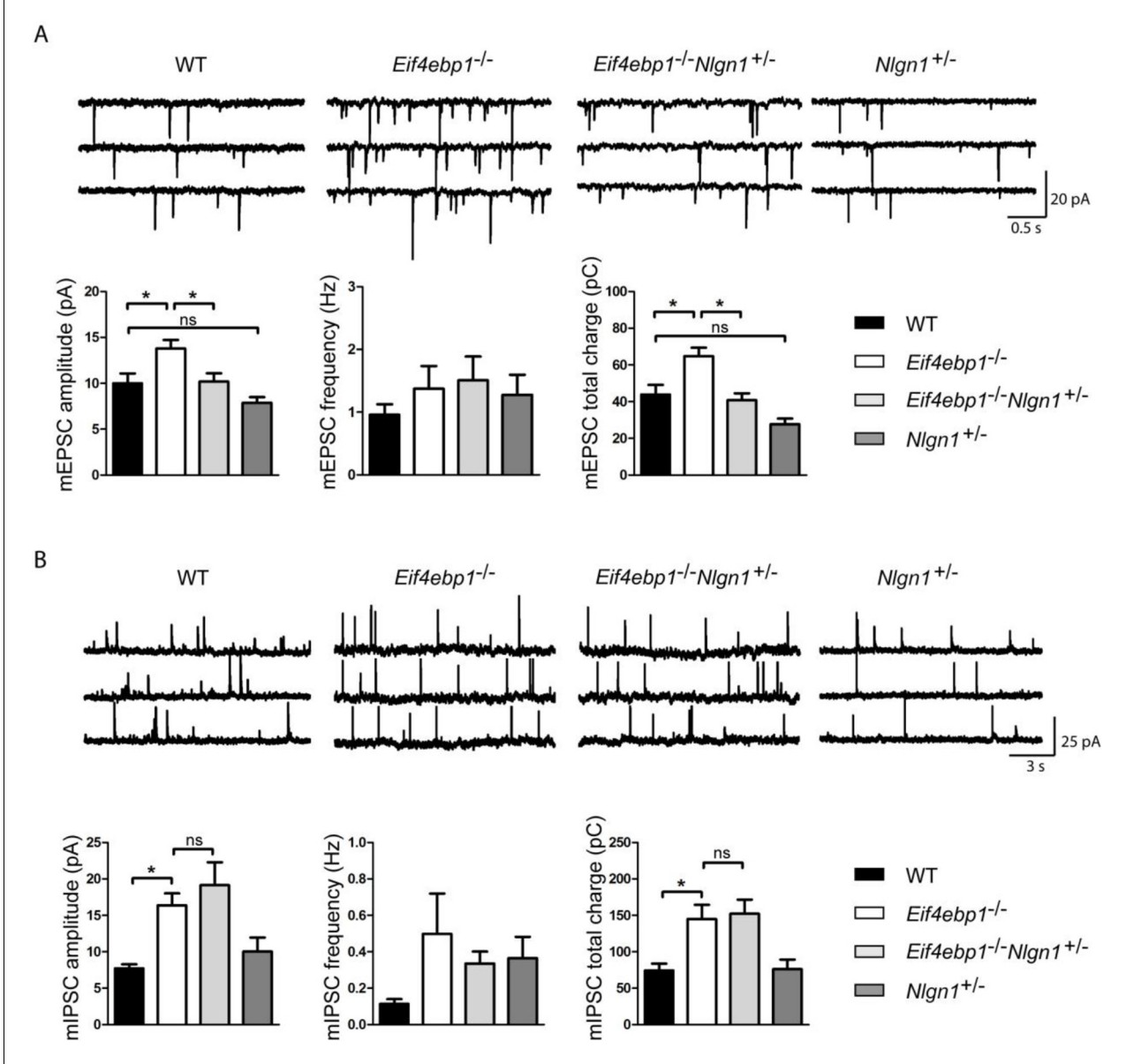

**Figure 5.** Reduction of neuroligin 1 levels in *Eif4ebp1*[-/-] mice normalizes the enhanced excitatory synaptic transmission. mEPSCs (**A**) and mIPSCs (**B**) were recorded from lamina II spinal neurons of wild-type (WT), *Eif4ebp1*[-/-], *Eif4ebp1*[-/-]/*Nlgn1*[+/-] and *Nlgn1*[+/-] mice. Top: representative traces of recordings from the four genotypes. Bottom: bar graph showing mEPSC (**A**) and mIPSC (**B**) amplitude, frequency and synaptic total charge transfer (n=11 WT; n=9 *Eif4ebp1*[-/-]; n=10 *Eif4ebp1*[-/-]/ *Nlgn1*[+/-]; n=7 Nlgn1[+/-]). The increase in mEPSC amplitude and total charge in *Eif4ebp1*[-/-] mice is rescued following deletion of one *Nlgn1* allele, whereas mIPSCs are not affected. Data are presented as mean ± SEM. *$p<0.05$ by Bonferroni post-hoc test following one-way ANOVA. See also figure supplement 1.

The following figure supplements are available for Figure 5:

**Figure supplement 1.** Input resistance and membrane capacitance are not altered in lamina II spinal neurons from *Eif4ebp1*[-/-], *Eif4ebp1*[-/-]/*Nlgn1*[+/-] and *Nlgn1*[+/-] mice.

therefore reasoned that the enhanced mechanical nociception in *Eif4ebp1*[-/-] mice could be mediated via enhanced translation of *Nlgn* mRNAs. Indeed, we found that the expression of neuroligins was increased in the spinal cord of *Eif4ebp1*[-/-] mice without concomitant changes in mRNA levels, indicating enhanced translation. The role of neuroligins in the spinal cord is not well studied. A recent study showed that knock-down of neuroligin 2, which promotes inhibitory synaptic transmission, in

the spinal cord of naïve rats increases mechanical sensitivity (*Dolique et al., 2013*). The role of neuroligin 1, which promotes excitatory synaptic transmission, in pain processing has not been investigated. Consistent with the higher levels of neuroligin 1, 2 and 3 in mice lacking 4E-BP1, we found that excitatory and inhibitory spinal synaptic transmission are increased in these animals. Chronic pain is associated with an increase in the excitatory synaptic drive to dorsal horn neurons (*Sandkuhler, 2009*). Therefore, in the current study we focused on neuroligin 1 and its role in promoting excitatory synaptic transmission in *Eif4ebp1$^{-/-}$* mice.

Interestingly, the enhancement of mEPSC amplitude and neuroligin 1 spinal expression in *Eif4ebp1$^{-/-}$* mice was rescued by intrathecal 4EGI-1. In agreement with these results, we also found that mechanical sensation is reduced in *Nlgn1$^{-/-}$* mice, indicating that neuroligin 1 has a pronociceptive function. Strikingly, normalization of neuroligin 1 levels rescued the increased mEPSC amplitudes in *Eif4ebp1$^{-/-}$* mice, and partially reversed the mechanical hypersensitivity in these mice. These data strongly indicate that augmented spinal expression of neuroligin 1 in *Eif4ebp1$^{-/-}$* mice contributes to mechanical hypersensitivity via neuroligin 1-induced upregulation of excitatory synaptic transmission in the spinal cord. The finding that normalization of neuroligin 1 levels only partially reverses mechanical hypersensitivity in *Eif4ebp1$^{-/-}$* mice indicates that additional mechanisms, peripheral or central, downstream of 4E-BP1/eIF4e contribute to the pain phenotype.

We observed that the increased inhibition in *Eif4ebp1$^{-/-}$* mice is solely mediated via enhanced glycinergic, but not GABAergic synaptic transmission (*Figure 3C*). Similarly to GABA$_A$, glycine receptors mediate fast synaptic inhibition in the spinal cord, where they regulate nociceptive and tactile sensory processing. Little is known about how synaptic strength of glycinergic synapses is regulated in vivo. Surprisingly, we found that the inhibitory synaptic transmission was not affected by intrathecal administration of 4EGI-1, possibly because of reduced sensitivity to 4EGI-1 or because inhibitory synaptic transmission is regulated by 4EGI-1 insensitive processes downstream of 4E-BP1. The differential regulation of glycinergic synaptic transmission by mTOR/4E-BP1 pathway in the spinal cord is intriguing, and further studies will be necessary to better understand the precise molecular mechanism of this regulation. However, the increased inhibition in *Eif4ebp1$^{-/-}$* mice was not reversed by normalization of neuroligin 1 expression (*Figure 5B*), and is thus unlikely to play major role in the reduced nociceptive thresholds of *Eif4ebp1$^{-/-}$* mice, which were reversed by normalization of neuroligin 1 expression (*Figure 4E,G*).

The threshold for enhancement of electrically-evoked field potentials is lowered in *Eif4ebp1$^{-/-}$* mice. This finding is consistent with studies in the hippocampus showing that 4E-BP2 ablation engenders the conversion of early-phase long-term potentiation (LTP) into late-phase LTP in the Schaffer collateral pathway (*Banko et al., 2005*). Lowered threshold for LTP induction is thought to be a mechanism contributing to increased responsiveness of spinal circuits under conditions of pathological pain (*Ji et al., 2003*; *Sandkuhler, 2009*). An mTORC1-mediated increase in eIF4F complex formation, which leads to the enhancement of synaptic transmission and lowering of the threshold for the induction of synaptic potentiation, is likely to contribute to abnormal pain processing in models of chronic pain. Though our study demonstrates an important role of enhanced expression of neuroligin 1 in the nociceptive phenotype of *Eif4ebp1$^{-/-}$* mice, it is conceivable that translational upregulation of other mRNAs contributes to the regulation of nociception downstream of mTORC1 and ribosome-profiling studies are required to identify these transcripts.

Our data demonstrate that the mTOR/4E-BP/eIF4E pathway controls mechanical, but not thermal sensitivity. A recent study showed that somatostatin-positive cells, which are enriched in lamina II of the dorsal horn where they comprise 53% of all glutamatergic excitatory neurons, are required for sensation of mechanical but not thermal pain (*Duan et al., 2014*). Interestingly, we found that lamina II neurons receive stronger excitatory input in *Eif4ebp1$^{-/-}$* mice (*Figure 3*), raising the intriguing possibility that the effect of 4E-BP1 depletion on mechanical, but not thermal sensation, could be mediated via increased excitatory drive to somatostatin-positive neurons. Conditional deletion of 4E-BP1 in somatostatin positive cells will be required to address this idea.

mTORC1 controls protein synthesis as well as lipid and ribosome biogenesis, autophagy and mitochondria function (*Costa-Mattioli and Monteggia, 2013*; *Shimobayashi and Hall, 2014*). Studies with rapalogues have shown that mTORC1 is strongly implicated in the development of hypersensitivity states in a variety of pain models (*Obara and Hunt, 2014*; *Price and Geranton, 2009*). However, whether the effects of mTORC1 on nociception are mediated via translation control mechanisms or via protein synthesis-independent functions of mTORC1 have remained largely unknown.

Our study clearly shows that translational regulators (4E-BP1 and eIF4F), acting downstream of mTOR, control nociception via their effect on mechanical sensation and synaptic responses.

In summary, our results shed important light on the mechanisms by which activation of the mTOR/4E-BP/eIF4E pathway enhances mechanical pain sensation. We show that 4E-BP1 is a major functional isoform in the pain pathway, where it represses translation of neuroligins in the spinal cord. Removal of 4E-BP1 increases neuroligin 1 expression, which in turn promotes excitatory synaptic transmission in the spinal cord and leads to mechanical hypersensitivity.

## Material and methods

### Mice

*Eif4ebp1, Eif4ebp2 and Eif4ebp1/2* null mutant mice were backcrossed for more than 10 generations into a C57BL/6J background (*Le Bacquer et al., 2007*). *Nlgn1* knockout mice were kindly provided by Craig M. Powell (The University of Texas Southwestern Medical Center, Dallas). Wild-type mice were littermates of the corresponding knockout mice. All behavioral experiments were performed on 8–12-week-old mice of both sexes by male experimenters blinded to genotype and drug. Food and water were provided *ad libitum*, and mice were kept on a 12:12 hr light/dark cycle (lights on at 08:00 hr). All procedures complied with the Canadian Council on Animal Care guidelines and International Association for the Study of Pain, and were approved by McGill University's Downtown Animal Care Committee.

### Behavioral assays

*von Frey Testing*: Mice were placed individually in transparent Plexiglas cubicles (5 × 8.5 × 6 cm) set on a perforated steel floor and habituated for 1 hr prior to testing. Nylon monofilaments (Stoelting #2-#9) were firmly applied to the plantar surface of each hind paw for 0.5 s. The up-down method of Dixon (*Chaplan et al., 1994*) was used to estimate the 50% withdrawal threshold (average of two measurements for both hind paws separated by at least 30 min).

*Tail Clip test*: A small alligator clip (700 g force) was applied at 1 cm from the base of the tail. The latency to attack/bite the clip was measured with a stopwatch to the nearest 0.1 sec. Upon attack, the clip was removed and the animals were returned to their cages.

*Radiant Heat Paw-Withdrawal*: Mice were placed in cubicles (described above) on a glass floor and a focused beam of high-intensity light was aimed at the plantar surface of the hind paw. The intensity of the light was set to 15% or 20% of maximum (IITC Model 390) with a cut-off value of 40 s. The latency to withdraw the hindpaw was measured to the nearest 0.1 s. Measurements consisted of testing both hind paws twice on two separate occasions separated by at least 30 min.

*Hot-plate test*: Mice were placed into a clear Plexiglas cylinder atop a metal surface (Columbus Instruments) maintained at 50°C. The latency to lick or shake either hind paw was measured with a stopwatch to the nearest 0.1 sec.

*Formalin test*: Mice were placed into Plexiglas cylinders on a glass floor and habituated for at least 30 min. Following habituation, all mice were given intraplantar injections of formalin (20 µl, 0.5%) into the left hind paw and placed back into the cylinders. Cameras, placed under the glass floor, recorded the licking behavior over 60 min. Video files were sampled at 1-min intervals for the presence or absence of licking behavior in the first 10 sec of each interval. Data are expressed as the percent of positive (licking) samples. The early phase of the formalin test was defined as 0–10 min. and the late phase as 10–60 min. post-injection.

### Electrophysiology

Electrophysiological recordings of miniature excitatory and inhibitory post-synaptic currents (mEPSCs and mIPSCS) and dorsal root-evoked field post-synaptic potentials (fPSPs) were made using lumbar spinal cord tissue. Adult (3−6 months old) male mice were anesthetized with urethane (2 g/kg) and perfused with ice-cold sucrose-substituted artificial cerebral spinal fluid (sucrose artificial cerebro-spinal fluid [aCSF]; contains in mM: 252 sucrose, 2.5 KCl, 1.5 $CaCl_2$, 6 $MgCl_2$, 10 d-glucose, bubbled with 95%:5% oxygen:$CO_2$). The lumbar spinal column was removed and immersed in ice-cold sucrose aCSF after which the spinal cord was quickly removed via laminectomy. For patch-clamp experiments, the dorsal and ventral roots were removed from the spinal cord and 300 µm-

thick parasagittal slices were cut from the lumbar portion in ice-cold sucrose aCSF. After cutting, the spinal cord slices were kept in oxygenated (95% $O_2$, 5% $CO_2$) aCSF (in mM: 126 NaCl, 2.5 KCl, 2 $MgCl_2$, 2 $CaCl_2$, 1.25 $NaH_2PO_4$, 26 $NaHCO_3$, 10 d-glucose) at room temperature until recording. For LTP experiments, the ventral roots and connective tissue were removed from the spinal cord after laminectomy, and the tissue explant was placed in room temperature aCSF for 1 hr before experimentation.

In most patch-clamp experiments, slices were continuously perfused (8 ml/min) with oxygenated aCSF supplemented with tetrodotoxin (1 μM). Lamina II cells were visually identified and patched with borosilicate pipettes containing (in mM): 110 $CsMeSO_3$, 11 Cs-EGTA, 10 CsCl, 20 HEPES, 2 $MgCl_2$, 1 $CaCl_2$, 4 Mg-ATP, 0.4 Tris-GTP, 0.1% (w/v) Lucifer Yellow; pH was adjusted to 7.2 with CsOH. Synaptic events were low-pass filtered at 2 kHz and recorded at 10 kHz using a Multiclamp 700B amplifier (Molecular Devices, Sunnyvale, CA), Digidata 1322A digitizer (Molecular Devices), and pClamp 10 software (Molecular Devices). DL-2-amino-5-phosphonovaleric acid (AP5; 40 μM) was added to the aCSF to block NMDA receptors. mEPSCs and mIPSCs were recorded at holding potentials of -70 mV and 0 mV, respectively. For experiments designed to isolate $GABA_A$ or glyciner-gic mIPSCs, strychnine (1 μM) or bicuculline (10 μM) were added to the aCSF, respectively, and visu-ally identified lamina II cells were patched with pipettes containing (in mM): 120 CsCl, 5 KCl, 11 EGTA, 1 $CaCl_2$, 4 Na-ATP, 0.4 Na-GTP, 10 HEPES and 0.1% Lucifer Yellow (wt/vol), pH 7.2. Stable 3 min recordings of mEPSCs and mIPSCs were selected for analysis in Clampfit 10 (Molecular Devices).

Electrically-evoked field potentials in the superficial dorsal horn were recorded as previously described (*Bonin and De Koninck, 2014*). fPSPs were recorded via a borosilicate glass electrode inserted into the dorsal side of the spinal cord explant in the dorsal root entry zone. Electrodes were inserted superficially to a depth of no more than 125 μm from the dorsal surface of the spinal cord measured with an MPC-200 manipulator (Sutter Instrument Company, Novato, CA, USA). Electrodes had a tip resistance of 3−4 MΩ when filled with aCSF. fPSPs were evoked by electrical stimulation of the dorsal root using a suction electrode that is pulled from borosilicate glass and filled with aCSF, and placed near the cut end of the dorsal root. Field potentials were amplified with a Multiclamp 700B amplifier (Molecular Devices, Sunnyvale, CA, USA), digitized with a Digidata 1322A (Molecular Devices), and recorded using pClamp 10 software (Molecular Devices). Data were filtered during acquisition with a low pass filter set at 1.6 kHz and sampled at 10 kHz. Test stimuli were presented every 60 s to evoke fPSPs and LTP was evoked by stimulation at 2 Hz for variable lengths of time. The stimulus intensity was sufficient to activate C-fibers, as indicated by the appearance of a third distinct fiber volley after the stimulus artifact (*Bonin and De Koninck, 2014*), while a slightly (10−20%) higher intensity was used to induce synaptic potentiation. Data were analyzed using ClampFit 10 software (Molecular Devices). The area of fPSPs relative to baseline was measured from 0−800 ms after the onset of the fPSP.

## Drugs

4EGI-1 was purchased from Calbiochem (San Diego, CA) and dissolved first in DMSO (stock solution) and then in 30% polyethylene glycol. All drugs and chemicals for electrophysiology experiments were purchased from Sigma-Aldrich (Oakville, ON, Canada).

## Western blotting

Tissue extracts for western blotting were prepared in ice-cold homogenization buffer containing (in mM): 20 Tris-HCl, pH 7.4; 150 NaCl; 1 EDTA; 1% Triton, 5 NaF; 1.5 $Na_3VO_4$, and protease inhibitor cocktail (complete, EDTA-free, Roche Applied Science). Synaptosomes were prepared using Syn-PER Synaptic Protein Extraction Reagent (Thermo Scientific CAT#8779. Following centrifugation at 12,000 × g for 10 min, the supernatant protein concentration was measured and samples containing equal protein amounts were boiled for 5 min in Laemmli sample buffer and separated by SDS-PAGE. Following electrophoresis, proteins were transferred to 0.2 μm nitrocellulose membranes. Membranes were blocked in 5% dry milk powder in Tris-buffered saline containing 0.1% Tween-20 (TBS-T) for 1 hr prior to overnight incubation with the primary antibody. The membranes were then washed, incubated for 1 h with HRP-conjugated secondary antibody, washed again, treated with Enhanced Chemiluminescence reagent (Perkin Elmer), and exposed to an autoradiography film

(Denville Scientific Inc.). All signals were obtained in the linear range for each antibody, quantified using ImageJ (NIH), and in some experiments normalized to β-actin. The western blot experiments were performed in triplicate. The antibodies and the dilutions for the western blots used are as follows: 4E-BP1 (1:1000, Cat#9644, Cell Signaling Technology), 4E-BP2 (1:1000, Cat#2845, Cell Signaling Technology), eIF4E (1:1000, Cat#610269, BD Transduction Laboratories), eIF4G1 (1:1000, Cat#ab2609 Abcam), neuroligin 1–3 (1:1000, Synaptic Systems), GluA1-2 (1:1000, Alomone labs) and β-actin (1:5000, Cat#A5441, Sigma).

## Measurement of de novo Pprotein sSynthesis

Eight-week-old wild-type and *Eif4ebp1*$^{-/-}$ mice were injected with puromycin (10 mg/kg, intraperitoneal, i.p.), and 45 min later DRGs were collected and processed for Western blotting, using anti-puromycin monoclonal antibody (3RH11, KeraFast). DRGs from mice injected with anisomycin (100 mg/kg i.p, injected 20 min before puromycin) or vehicle were processed in parallel and used as controls. Protein synthesis was determined by measuring total lane signal from 250–15 KDa and subtracting unlabelled protein control. Signals were quantified using ImageJ.

## Immunohistochemistry

Mice were deeply anaesthetized and transcardially perfused with 30 ml vascular rinse (0.1% w/v sodium nitrite in 0.01M perfusion buffer) (for composition see Côté et al, 1993) and 200 ml fixative solution (4% paraformaldehyde in 0.1 M PB, pH 7.4) at room temperature (RT). Spinal cords were extracted trough laminectomy and post-fixed for 2 h at 4°C. Lumbar levels (L3-L5) of spinal cords were sectioned using a vibratome and collected in PB saline (PBS) (50-μm thick free floating transverse sections). Sections were washed in PBS containing 0.2% Triton X-100 (PBS-T; Sigma Aldrich) for 30 min at RT. To block unspecific staining, sections were treated with PBS-T containing 10% normal goat serum (NGS) for 1 hr at RT. Sections were then incubated with primary antibodies diluted in PBS-T containing 5% NGS, overnight at 4°C. Sections were washed in PBS-T for 30 min and incubated with secondary antibodies diluted 1:800 in PBS-T, at RT for 2 hr, protected from light. For isolectin B4 (IB4) staining, sections were incubated with IB4 conjugated to AlexaFluor 546, 1:200 (Molecular Probes) for 1 hr at RT. All steps were carried out under constant shaking. After 30 min of final washes in PBS-T and 10 min in PBS, sections were mounted with SlowFade Gold (Invitrogen, Burlington, ON, Canada). The slides were stored protected from light at 4°C until examined by use of the Olympus confocal microscope using a 20x oil-immersion objective. Antibodies used for immunohistochemistry were: glial fibrillary acidic protein (GFAP, 1:2000, GFAP Mouse mAb, Cell Signalling, #3670), NEUronal Nuclei (NeuN, 1:5000, Mouse Anti-NeuN Antibody, clone A60, MAB377), calcitonin gene-related peptide (CGRP, 1:1000, Sigma-Aldrich, C8198), substance P (1:400, Neuromics, MO15094), HTR3A (1:200, Sigma-Aldrich, AV13046), and c-Fos (1:10000, Calbiochem, PC38). For detection, species-specific secondary antibodies were used: goat anti-rabbit Alexa-Fluor 488 (1:800, Invitrogen Molecular probes) and goat anti-mouse Alexa-Fluor 546 (1:800, Invitrogen Molecular Probes).

## Lentivirus packaging and spinal cord injection

Lentiviral vectors for shRNA silencing of 4E-BP1 and a scrambled sequence were obtained from Sigma. The Sigma MISSION shRNA vectors accession numbers were: mouse 4E-BP1 (TRCN0000075612), and the Non-Target shRNA Control (SHC002). Each shRNA vector was co-transfected into HEK293T cells with the lentivirus packaging plasmids PLP1, PLP2, and PLP-VSVG (Invitrogen) using Lipofectamine 2000 (Invitrogen). Viral supernatants were collected 48 and 72 h post-transfection. Supernatants were filtered through a 0.45 μm nitrocellulose filter either before being applied to target cells in the presence of polybrene (5 μg/ml), or processed for concentration by ultracentrifugation at 26,000 rpm/2 h/4°C with 20% sucrose cushion. Virus concentrates were resuspended in serum free media, and aliquots were stored at -80°C until use. Viral titre (TU/ml) was calculated using puromycin selection of MEFs infected with different viral particle dilutions and stained with 0.2% crystal violet and 20% methanol solution. Colonies were measured using the CellCount plugin of ImageJ (NIH). Viral titres were adjusted to 1.5 10$^7$ TU/ml for in vivo injections.

For intraparenchymal spinal cord dorsal horn (SCDH) injection, mice were anaesthetized with an i.p. injection of a cocktail containing xylazine (3.33 mg/ml), ketamine (55.55 mg/ml), and Domitor

(0.27 mg/ml). The mouse was placed in a spinal frame and a laminectomy was used to remove spinous process VL3 and VL4. Six unilateral injections of 0.25 μl ($1 \times 10^7$ viral particles/ml) were administesred 0.7 mm apart, at a depth of 0.2 mm (SCDH), using a glass pipette attached to a Hamilton syringe with plastic tubing at a rate of 0.1 μl/min. To allow for the solution containing viral particles to diffuse from the tip of the glass pipette into the tissue, the pipette stayed in the tissue for one additional minute. The syringe was mounted on a microinjector (Nanomite, Harvard Apparatus) attached to a stereotaxic unit (David Kopf Instruments). The mice were allowed to recover for 1 week before the experiments.

## m7GDP pull-down

Lumbar spinal cord was homogenized in lysis buffer containing 40 mM HEPES-KOH (pH 7.5), 120 mM NaCl, 1 mM EDTA, 0.1 mM GDP, 10 mM pyrophosphate, 10 mM β-glycerophosphate, and 50 mM NaF. Extracts were mixed with 30 μl of $m^7$GDP-agarose for 1.5 h at 4°C. The resins were washed four times and proteins were eluted with SDS–PAGE sample buffer.

## Statistical Analyses

All results are expressed as mean ± SEM. All statistical comparisons were made with either Student's *t*-test or one-way or two-way ANOVA, followed by between-group comparisons using *t*-test or Bonferroni's post-hoc test, unless otherwise indicated, with $p<0.05$ as the significance criteria.

## Acknowledgments

This work was supported by a CIHR operating grant to N. Sonenberg (MOP-114994), an unrestricted gift from the Louise and Alan Edwards Foundation to JS Mogil, a CIHR grant to Y. De Koninck (MOP-12942) and the Louise and Alan Edwards Foundation fellowship to R. Bonin. We thank A Sylvestre, S Perreault, C Lister, and I Harvey for technical assistance.

## Additional information

### Competing interests

NS: Reviewing editor, *eLife.* The other authors declare that no competing interests exist.

### Funding

| Funder | Grant reference number | Author |
| --- | --- | --- |
| Canadian Institutes of Health Research | Operating grant to N. Sonenberg (MOP-114994) | Arkady Khoutorsky<br>Christos Gkogkas<br>Seyed Mehdi Jafarnejad<br>Tommy Alain<br>Nahum Sonenberg |
| The Louise and Alan Edwards Foundation | An unrestricted gift to J.S. Mogil | Robert E Sorge<br>Loren Martin<br>Jeffrey S Mogil |

The funders had no role in study design, data collection and interpretation, or the decision to submit the work for publication.

### Author contributions

AK, RPB, RES, JPS, Conception and design, Acquisition of data, Analysis and interpretation of data, Drafting or revising the article; CGG, JSM, NS, Conception and design, Analysis and interpretation of data, Drafting or revising the article; SAP, SMJ, ARda-S, YDK, FC, Conception and design, Drafting or revising the article; MHP, EWS, Acquisition of data, Analysis and interpretation of data, Drafting or revising the article; TA, Drafting or revising the article, Contributed unpublished essential data or reagents; LM, Acquisition of data, Drafting or revising the article

### Author ORCIDs

Robert P Bonin, http://orcid.org/0000-0002-3287-6803
Yves De Koninck, http://orcid.org/0000-0002-5779-9330

## Ethics

Animal experimentation: All procedures complied with the Canadian Council on Animal Care guidelines and International Association for the Study of Pain, and were approved by McGill University's Downtown Animal Care Committee.

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
