## [Decision Letter]

[Editors’ note: a previous version of this study was rejected after peer review, but the authors submitted for reconsideration. The first decision letter after peer review is shown below.]

Thank you for choosing to send your work entitled "Translational control of nociception via 4E-binding protein 1" for consideration at *eLife*. Your full submission has been evaluated by a Senior editor, a Reviewing editor, and two reviewers, and the decision was reached after discussions between the reviewers. We regret to inform you that your work will not be considered further for publication in *eLife* in its current form. However, if you can address the concerns we would be happy to consider a new submission at a later date with no guarantee of acceptance.

The main concerns with the manuscript are the lack of evidence to show if there is any reorganization in the dorsal horn of the knockout mouse and whether the viral delivery system really does target dorsal horn neurons and not just the dorsal roots/DRGs.

Khoutorsky et al. show that deletion of the eukaryotic initiation factor 4E-binding protein 1 (4E-BP1), an mTOR effector leads to mechanical, but not thermal pain hypersensitivity. Mice lacking 4E-BP1 show increased excitatory and inhibitory synaptic input into spinal neurons, and a lowered threshold for induction of synaptic potentiation. They also show enhanced translation of neuroligin 1, a cell-adhesion postsynaptic protein. Pharmacological inhibition of eIF4E activity or genetic reduction of neuroligin 1 levels normalizes the increased excitatory synaptic input and reverses mechanical hypersensitivity. They show that translational control by 4E-BP1 downstream of mTOR regulates the expression of neuroligin 1 and excitatory synaptic transmission in the spinal cord and contributes to enhanced mechanical nociception. The manuscript is well written and the results are presented clearly.

1) Throughout the manuscript the authors use viral infection to raise or lower various levels of protein translation in the dorsal horn but it is generally acknowledged (see for example Simonetti, M. M. et al., 2013, Neuron, 77(1), 43-57) that virus will not enter the dorsal horn when injected intrathecally and must be injected directly into the spinal cord. In fact Levine (see Bogen, O. O. et al., 2012, The Journal of Neuroscience 32(6), 2018-2026 and others) have often used this route to infect DRGs. This is important because it is unclear from the manuscript if most of their manipulations are actually on the dorsal root or DRG particularly that local translation seems to have been demonstrated in primary afferent fibers (Price, T. J. et al., 2007, Journal of Neuroscience, 27(51), 13958-13967) and particularly in A-nociceptors (Jiménez-Díaz, L. et al., 2008, PLoS ONE, 3(4), e1961. As I understand it, the reduced function of A-nociceptors is in fact the explanation Geranton, S. M. et al. (2009, Journal of Neuroscience, 29(47), 15017-15027)) offer for the reduction of mechanical sensitivity following i.t. rapamycin. Could this explain many of the results here? 4EBP has been shown to be present in peripheral nerves (Melemedjian, O. K. et al., 2011, Molecular Pain, 7(1), 70. doi:10.1186/1744-8069-7-70; Jiménez-Díaz, L. et al., 2008, PLoS ONE, 3(4), e1961) even though a single report did not show a great deal of protein in DRG neurons. Also rapamycin was shown by various groups only to reduce mechanical hypersensitivity (but see Norsted Gregory, E. et al., 2010, Neuroscience, 169(3), 1392-1402). The authors need additional immunohistochemical and/or functional experiments to determine where the virus is functioning.

2) The mouse is a global knockout and it is therefore desirable to know whether there is any reorganization within the dorsal horn. For example, C-fiber termination, lamina markers and descending pathways such as 5HT and also glial markers should be mapped. If formalin was being tested then c-fos activation or pERK activation should also have been compared between genotypes would be a welcome addition.

3) The ephysiology suggests there is an increase in excitability and inhibition within the dorsal horn. The authors should examine the contributions of both A and C fibers by differential stimulation and conduction velocity measurements (see Torsney, C., and MacDermott, A. B., 2006, Journal of Neuroscience, 26(6), 1833-1843) and on identified neurons either in lamina I or II. This would indicate whether the mechanical sensitivity is driven by both C and/ or Ad fibers.

4) The authors show that the eIF4E inhibitor 4EGI-1 can reverse the mechanical pain hypersensitivity, but they did not determine whether it normalized the increase in mEPSC amplitude in the 4E-BP1 knockout mice. The authors should determine whether 4EGI-1 can normalize the mEPSC amplitude in the 4E-BP1 knockout mice. Similarly, the authors should determine whether 4EGI-1 normalizes the increase in the levels of the neuroligins. Related to this, the authors suggest that increased neuroligin2 levels in the 4E-BP1 are likely responsible for the increase in mIPSC amplitude and frequency. Although I don't think it is necessary to directly determine whether this is the case (although it would strengthen the model), the authors should determine whether 4EGI-1 normalizes the increase in mIPSC amplitude and frequency in the 4E-BP1 knockout mice.

*Reviewer #1*:

Khoutorsky et al show that deletion of the eukaryotic initiation factor 4E-binding protein 1 (4E-BP1), an mTOR effector leads to mechanical, but not thermal pain hypersensitivity. Mice lacking 4E-BP1 show increased excitatory and inhibitory synaptic input into spinal neurons, and a lowered threshold for induction of synaptic potentiation. They also show enhanced translation of neuroligin 1, a cell-adhesion postsynaptic protein. Pharmacological inhibition of eIF4E activity or genetic reduction of neuroligin 1 levels normalizes the increased excitatory synaptic input and reverses mechanical hypersensitivity. They show that translational control by 4E-BP1 downstream of mTOR regulates the expression of neuroligin 1 and excitatory synaptic transmission in the spinal cord and contributes to enhanced mechanical nociception. The research is led by Nahum Sonnenberg who is one of the acknowledged leaders in the field of the molecular biology of translation and obviously this aspect is sound. I am less convinced by the analysis of pain behavior and the efficacy of some of the techniques used here which raise substantial problems of interpretation.

Throughout the manuscript the authors use viral infection to raise or lower various levels of protein translation in the dorsal horn but it is generally acknowledged (see for example Simonetti, M. M. et al., 2013, Neuron, 77(1), 43-57) that virus will not enter the dorsal horn when injected intrathecally and must be injected directly into the spinal cord. In fact Levine (see Bogen, O. O. et al., 2012, The Journal of Neuroscience 32(6), 2018-2026 and others) have often used this route to infect DRGs. This is important because it is unclear from the manuscript if most of their manipulations are actually on the dorsal root or DRG particularly that local translation seems to have been demonstrated in primary afferent fibers (Price, T. J. et al., 2007, Journal of Neuroscience, 27(51), 13958-13967) and particularly in A-nociceptors (Jiménez-Díaz, L. et al., 2008, PLoS ONE, 3(4), e1961. As I understand it, the reduced function of A-nociceptors is in fact the explanation Geranton, S. M. et al. (2009, Journal of Neuroscience, 29(47), 15017-15027)) offer for the reduction of mechanical sensitivity following i.t. rapamycin. Could this explain many of the results here? 4EBP has been shown to be present in peripheral nerves (Melemedjian, O. K. et al., 2011, Molecular Pain, 7(1), 70. doi:10.1186/1744-8069-7-70; Jiménez-Díaz, L. et al., 2008, PLoS ONE, 3(4), e1961) even though a single report did not show a great deal of protein in DRG neurons. Also rapamycin was shown by various groups only to reduce mechanical hypersensitivity (but see Norsted Gregory, E. et al., 2010, Neuroscience, 169(3), 1392-1402). The authors need additional immunohistochemical and/or functional experiments to determine where the virus is functioning.

The authors need to show where the virus goes with immunohistochemical approaches. It would make life simpler for all of us if dorsal horn could be infected this way but I suspect this is not the case.

The authors describe mechanical hypersensitivity and an increased formalin response. I am not clear why inflammation or capsaicin induced thermal and mechanical sensitivity were not tested, or indeed a neuropathic or surgical pain model. This may discount any effect on C fibers at least. The pain scientists listed on this manuscript have used these approaches many times before.

The mouse is also a global knockout and it is usual to show that there is no major reorganization within the dorsal horn. For example, C-fiber termination, lamina markers and descending pathways such as 5HT and also glial markers should be mapped. If formalin was being tested then c-fos activation or pERK activation should also have been compared between genotypes.

The ephysiology suggests there is an increase in excitability and inhibition within the dorsal horn. The authors should examine the contributions of both A and C fibers by differential stimulation and conduction velocity measurements (see Torsney, C., and MacDermott, A. B., 2006, Journal of Neuroscience, 26(6), 1833-1843) and on identified neurons either in lamina I or II. This would indicate whether the mechanical sensitivity is driven by both C and/ or Ad fibers.

Was there a thermal deficit in the *Nlgn1^-/-^* mouse?

In summary the authors need to add convincing data showing that there is no substantial reorganization in the dorsal horn of the knockout mouse and that viral interventions really do effect the dorsal horn and not just the dorsal roots/DRGs (-and it is likely they must influence DRGs and dorsal roots). Finally the physiology needs to distinguish between C and Ad fibers and any changes in their excitability (perhaps with CAPs) and also the postsynaptic consequences of their activity in the dorsal horn of the mutant and littermate control mouse. There are also notable omissions in the reference list.

*Reviewer #2*:

Herein Khoutorsky et al. have conducted experiments examining the role of eIF4E-dependent translation in nociception by examining mechanical and thermal pain sensitivity in mice lacking the repressor 4E-BP1. The authors found that 4E-BP1 knockout mice display 1) mechanical (but not thermal) pain hypersensitivity, 2) increased excitatory mEPSC amplitude in spinal cord neurons, 3) increased inhibitory mIPSC amplitude and frequency in spinal cord neurons, and 4) increased levels of neuroligin1, neuroligin 2, and neuroligin 3. Finallly, the increase in mEPSC amplitude was normalized when the 4E-BP1 knockout mice were crossed with neuroligin1 heterozygous knockout mice.

The manuscript is well written, the results are presented clearly, and the findings are novel. However, I think that there a couple of experimental gaps that authors should fill to make their model more convincing.

1) The authors show that the eIF4E inhibitor 4EGI-1 can reverse the mechanical pain hypersensitivity, but they did not determine whether it normalized the increase in mEPSC amplitude in the 4E-BP1 knockout mice. The authors should determine whether 4EGI-1 can normalize the mEPSC amplitude in the 4E-BP1 knockout mice.

2) Similarly, the authors should determine whether 4EGI-1 normalizes the increase in the levels of the neuroligins.

3) Finally, the authors suggest that increased neuroligin2 levels in the 4E-BP1 are likely responsible for the increase in mIPSC amplitude and frequency. Although I don't think it is necessary to directly determine whether this is the case (although it would strengthen the model), the authors should determine whether 4EGI-1 normalizes the increase in mIPSC amplitude and frequency in the 4E-BP1 knockout mice.

---

## [Author Response]

[Editors’ note: the author responses to the first round of peer review follow.]

In response to the reviewers’ requests we have performed additional experiments and the new data are now included in the revised manuscript (Figure 1, Figure 3, Figure 4, and Figure 1—figure supplement 2, Figure 1—figure supplement 3). We are gratified by the positive comments: “the manuscript is well written, the results are presented clearly, and the findings are novel” and “the main concerns with the manuscript are the lack of evidence to show if there is any reorganization in the dorsal horn of the knockout mouse and whether the viral delivery system really does target dorsal horn neurons and not just the dorsal roots/DRGs.” To address these major concerns and other issues raised by the reviewers we have performed additional experiments as detailed below.

*1) Throughout the manuscript the authors use viral infection to raise or lower various levels of protein translation in the dorsal horn but it is generally acknowledged (see for example Simonetti, M. M. et al., 2013, Neuron, 77(1), 43-57) that virus will not enter the dorsal horn when injected intrathecally and must be injected directly into the spinal cord. In fact Levine (see Bogen, O. O. et al., 2012, The Journal of Neuroscience 32(6), 2018-2026 and others) have often used this route to infect DRGs. This is important because it is unclear from the manuscript if most of their manipulations are actually on the dorsal root or DRG particularly that local translation seems to have been demonstrated in primary afferent fibers (Price, T. J. et al., 2007, Journal of Neuroscience, 27(51), 13958-13967) and particularly in A-nociceptors (Jiménez-Díaz, L. et al., 2008, PLoS ONE, 3(4), e1961. As I understand it, the reduced function of A-nociceptors is in fact the explanation Geranton, S. M. et al. (2009, Journal of Neuroscience, 29(47), 15017-15027)) offer for the reduction of mechanical sensitivity following i.t. rapamycin. Could this explain many of the results here? 4EBP has been shown to be present in peripheral nerves (Melemedjian, O. K. et al., 2011, Molecular Pain, 7(1), 70. doi:10.1186/1744-8069-7-70; Jiménez-Díaz, L. et al., 2008, PLoS ONE, 3(4), e1961) even though a single report did not show a great deal of protein in DRG neurons. Also rapamycin was shown by various groups only to reduce mechanical hypersensitivity (but see Norsted Gregory, E. et al., 2010, Neuroscience, 169(3), 1392-1402). The authors need additional immunohistochemical and/or functional experiments to determine where the virus is functioning.*

Reviewer #1 raised a concern that intrathecal delivery of viral particles does not allow to distinguish between the effect of 4E-BP1 knockdown in DRGs and spinal cord. We agree that intraparenchymal spinal cord dorsal horn injection is a better approach to selectively reduce the expression levels of 4E-BP1 in the spinal cord, and therefore we followed the reviewer’s advice and injected viral particles intraparenchymally into the dorsal horn of the spinal cord (Figure 1). To improve the efficiency of 4E-BP1 knockdown the injections were performed at six locations along the lumbar segment of the spinal cord (see experimental procedures). We showed that the virus-driven eGFP expression is restricted to the dorsal horn and exhibits mostly neuronal distribution as it co-localizes with a neuronal marker NeuN (Figure 1—figure supplement 3). We demonstrated that the levels of 4E-BP1 were reduced in the superficial spinal cord but not in DRGs. Using this approach we substantiate our findings obtained in *Eif4ebp1^-/-^* mice to show that downregulation of 4E-BP1 specifically in the spinal cord induces mechanical hypersensitivity (Figure 1).

The reviewer commented that “*the reduced function of A-nociceptors is in fact the explanation Geranton, S. M. et al. (2009, Journal of Neuroscience, 29(47), 15017-15027)) offer for the reduction of mechanical sensitivity following i.t. rapamycin*.” And “*Also rapamycin was shown by various groups only to reduce mechanical hypersensitivity (but see Norsted Gregory, E. et al., 2010, Neuroscience, 169(3), 1392-1402)*.”

A direct comparison between the effects of rapamycin and modulation of eIF4F activity (by 4EGI-1 or 4E-BP1 downregulation) is not warranted because of the following reasons:

1) mTORC1 has two main downstream effectors, S6Ks and 4E-BPs. Rapamycin inhibits mTORC1-mediated phosphorylation of S6Ks more dramatically than that of 4E-BPs (Choo and Blenis, 2009). Therefore the effects of rapamycin on cellular functions are mediated to a greater extent via inhibition of S6Ks as compared to 4E-BPs. By selective targeting of 4E-BP1, our study provides unique insights into the mechanisms by which mTOR regulates nociception via 4E-BPs.

2) mTORC1 controls protein synthesis as well as lipid and ribosome biogenesis, autophagy and mitochondria function (Costa-Mattioli and Monteggia, 2013; Shimobayashi and Hall, 2014). These translation independent functions of mTORC1 have been recently implicated in many physiological and pathological processes in neurons (Malik et al., 2013; Tang et al., 2014). Because rapamycin affects both mRNA translation and other processes downstream of mTORC1, it is difficult to conclude whether the effects of rapamycin on nociception are mediated via mRNA translation or other mechanisms. Thus, it is imperative to investigate specific roles of 4E-BPs and eIF4F complex in modulation of mechanical hypersensitivity. A previous study showed that intraplantar injection of IL-6 and NGF upregulates eIF4F complex formation and cap-dependent translation in primary afferent neurons, and induces mechanical allodynia, which is completely blocked by co-injection of 4EGI-1 (Melemedjian et al., 2010). These results demonstrate that eIF4F complex formation is critical for IL-6 and NGF-mediated sensitization of sensory neurons. However, the roles of eIF4F and its main regulator 4E-BP1 in sensitization of spinal circuits remained unknown.

Reviewer 1 writes “*As I understand it, the reduced function of A-nociceptors is in fact the explanation Geranton, S. M. et al. (2009, Journal of Neuroscience, 29(47), 15017-15027)) offer for the reduction of mechanical sensitivity following i.t. rapamycin*.”

We thank the reviewer for raising this important issue and we apologize that we were not clear in the first submission. The paper from Stephen Hunt’s group demonstrates that rapamycin increases the thermal and electrical threshold of activation of A-fibers. However, it also shows that “The number of P-S6 protein expressing cells increases in the superficial dorsal horn after capsaicin administration, especially in lamina I, and rapamycin can attenuate this effect”, concluding that “intrathecal rapamycin has effects at both sites, resulting in a profound reduction in neuropathic pain sensitivity.”

We do not exclude the possibility that primary sensory neurons contribute to the pain phenotypes of *Eif4ebp1^-/-^* mice, however in the current study we specifically focus on spinal mechanisms by which 4E-BP1 enhances mechanical sensation. In our work we selectively downregulate 4E-BP1 in the spinal cord dorsal horn and manipulate the expression level of neuroligin 1, which controls synaptic transmission. To assess synaptic functions we measure mEPSC/mIPSC amplitude and frequency, and activity-dependent synaptic potentiation. We find that while normalization of neuroligin 1 levels in *Eif4ebp1^-/-^* mice corrects increased excitatory synaptic inputs to spinal neurons, it only partially reverses the enhanced mechanical sensation. These results are consistent with the existence of other 4E-BP1-dependent mechanisms. In the revised version we explain better that the current study characterizes the mechanisms by which 4E-BP1 regulates nociception in the spinal cord (paragraph three, subsection “4E-BP1 ablation induces mechanical hypersensitivity” and Discussion). Additionally, we extend the description of the literature on the role of mTOR and eIF4E in sensitization of primary sensory neurons (paragraph three, Introduction and Discussion). *2) The mouse is a global knockout and it is therefore desirable to know whether there is any reorganization within the dorsal horn.*

We thank the reviewer for this important comment. We have now included in the revised version a characterization of the dorsal horn of the spinal cord of *Eif4ebp1^-/-^* mice using neuronal marker NeuN, peptidergic and non-peptidergic C fiber markers (SP and CGRP for peptidergic, and IB4 for non-peptidergic), glial marker (GFAP) and 5-HT receptor immunostaining (HTR3A, Figure 1—figure supplement 2). We found no gross macroscopic alterations in the organization of the dorsal horn of *Eif4ebp1^-/-^* mice.

*If formalin was being tested then c-fos activation or pERK activation should also have been compared between genotypes would be a welcome addition.*

We now show that intraplantar formalin injection induces greater c-Fos expression (at the 2-hour time point) in the dorsal horn of *Eif4ebp1^-/-^* as compared to WT mice (Figure 1).

*3) The ephysiology suggests there is an increase in excitability and inhibition within the dorsal horn. The authors should examine the contributions of both A and C fibers by differential stimulation and conduction velocity measurements (see Torsney, C., and MacDermott, A. B., 2006, Journal of Neuroscience, 26(6), 1833-1843) and on identified neurons either in lamina I or II.*

We think that the increased synaptic transmission in lamina II neurons of *Eif4ebp1^-/-^* is a result of enhanced excitability of spinal circuits and not of primary afferents. This is supported by the fact that spontaneous synaptic transmission, which is independent of presynaptic neuron evoked activity, is enhanced in *Eif4ebp1^-/-^* mice and can be rescued by neuroligin 1 downregulation (Figure 5). Moreover, the enhanced inhibition must necessarily also arise from intraspinal circuits since sensory afferents are not inhibitory. While we feel that investigating whether the mechanical sensitivity of *Eif4ebp1^-/-^* mice is driven by C and/ or Aδ fibers should be addressed in future studies, we made significant efforts to measure the contribution of C and A fibers, as suggested by the reviewer. It is noteworthy that the Torsney and MacDermott paper, which the reviewer refers to, uses different approaches than used here. They used a rat preparation, which allows for the preservation of much longer sensory afferents for enhanced temporal separation between A- and C-fiber input than is generally possible with mice, resulting in reduced response noise. The Torsney and MacDermott study is also restricted to NK1+ neurons, which is a much more homogeneous neuron population than that studied here, and which is largely absent from lamina II, where we focused our efforts. A detailed study of all defined cell populations in the dorsal horn is a long term commitment. We performed a lengthy investigation of afferent-evoked responses in lamina II neurons to permit extrapolation from our reported electrophysiological results. We consistently observed an overly large degree of variability in these data that did not permit conclusive interpretation. Although this is not entirely unexpected given the heterogeneity between lamina II neurons and their likely synaptic input, we feel that the inherent difficulties of this approach further support our focus on spinal mechanisms of 4E-BP1 inhibition: the contrast of the heterogeneity of evoked responses in lamina II neurons with the consistent effect of 4E-BP1 modulation on miniature synaptic activity in lamina II neurons reinforces our position (which is supported by numerous, consistent lines of evidence in this manuscript) that changes in pain behaviour produced by 4E-BP1 inhibition are likely related to an increase in excitability of spinal cord sensory processing pathways.*4) The authors should determine whether 4EGI-1 normalizes the increase in mIPSC amplitude and frequency in the 4E-BP1 knockout mice.*

In the revised submission we included a series of experiments directly addressing this request.

We show that the regimen of 4EGI-1 administration (10 μg, i.t. daily for 3 days) that we used to rescue the mechanical hypersensitivity of *Eif4ebp1^-/-^* mice normalizes the increased mEPSC amplitude in lamina II *Eif4ebp1^-/-^* neurons (Figure 3). 4EGI-1 treatment also normalizes the enhanced expression of neuroligin 1 in the spinal cord. However, unexpectedly, we find that the enhanced inhibitory synaptic transmission in *Eif4ebp1^-/-^* neurons is not rescued by 4EGI-1. In the Discussion section we propose several explanations for the lack of effect of 4EGI-1 on inhibitory synaptic transmission.

Reviewer #1:*[…] I am not clear why inflammation or capsaicin induced thermal and mechanical sensitivity were not tested, or indeed a neuropathic or surgical pain model.*

We have tested both chronic inflammation-induced pain (in CFA model), and neuropathic pain (in CCI model) in *Eif4ebp1^-/-^* mice (Figure 6, Figure 7).

Author response image 1.**DOI:**
http://dx.doi.org/10.7554/eLife.12002.012

Our data show no differences in thermal hyperalgesia in CFA model. In CFA and neuropathic CCI models, both WT and *Eif4ebp1^-/-^* mice develop mechanical allodynia, however since at the baseline prior to CFA injection or surgeries the mechanical sensitivity is greatly increased in *Eif4ebp1^-/-^* mice, we feel that this constitutes a confounding factor that greatly complicates making meaningful conclusions regarding inflammation/injury-induced allodynia. Therefore, we hesitate to include these data in the manuscript. However, if the editor and reviewers deem it necessary to include these results, we will incorporate them into the new version with the caveat described above.

Author response image 2.**DOI:**
http://dx.doi.org/10.7554/eLife.12002.013

*There are also notable omissions in the reference list.*

We have modified and extended the reference list in the revised version.

References:

Choo, A.Y., and Blenis, J. (2009). Not all substrates are treated equally: implications for mTOR, rapamycin-resistance and cancer therapy. Cell Cycle 8, 567-572.

Malik, A.R., Urbanska, M., Macias, M., Skalecka, A., and Jaworski, J. (2013). Beyond control of protein translation: what we have learned about the non-canonical regulation and function of mammalian target of rapamycin (mTOR). Biochimica et biophysica acta 1834, 1434-1448.

Melemedjian, O.K., Asiedu, M.N., Tillu, D.V., Peebles, K.A., Yan, J., Ertz, N., Dussor, G.O., and Price, T.J. (2010). IL-6- and NGF-induced rapid control of protein synthesis and nociceptive plasticity via convergent signaling to the eIF4F complex. The Journal of neuroscience: the official journal of the Society for Neuroscience 30, 15113-15123.

Tang, G., Gudsnuk, K., Kuo, S.H., Cotrina, M.L., Rosoklija, G., Sosunov, A., Sonders, M.S., Kanter, E., Castagna, C., Yamamoto, A., et al. (2014). Loss of mTOR-dependent macroautophagy causes autistic-like synaptic pruning deficits. Neuron 83, 1131-1143.